# Chronic Inflammation as the Underlying Mechanism of the Development of Lung Diseases in Psoriasis: A Systematic Review

**DOI:** 10.3390/ijms23031767

**Published:** 2022-02-04

**Authors:** Mateusz Mleczko, Agnieszka Gerkowicz, Dorota Krasowska

**Affiliations:** Department of Dermatology, Venerology and Pediatric Dermatology, Medical University of Lublin, 20-081 Lublin, Poland; agerkowicz@wp.pl (A.G.); dor.krasowska@gmail.com (D.K.)

**Keywords:** asthma, chronic obstructive pulmonary disease, obstructive sleep apnea, pulmonary hypertension, interstitial lung disease, sarcoidosis, psoriasis, systematic review

## Abstract

Psoriasis is a systemic inflammatory disease caused by dysfunctional interactions between the innate and adaptive immune responses. The systemic inflammation in psoriasis may be associated with the development of comorbidities, including lung diseases. In this review, we aimed to provide a summary of the evidence regarding the prevalence of lung diseases in patients with psoriasis and the potential underlying mechanisms. Twenty-three articles published between March 2010 and June 2021 were selected from 195 initially identified records. The findings are discussed in terms of the prevalence of asthma, chronic obstructive pulmonary disease, interstitial lung disease, obstructive sleep apnea, pulmonary hypertension, and sarcoidosis in psoriasis. A higher prevalence of lung diseases in psoriasis has been confirmed in asthma, chronic obstructive pulmonary disease, obstructive sleep apnea, and pulmonary hypertension. These conditions are important as they are previously unrecognized causes of morbidity and mortality in psoriasis. The development of lung diseases in patients with psoriasis can be explained by several mechanisms, including common risk factors, shared immune and molecular characteristics associated with chronic inflammation, as well as other mechanisms. Understanding the prevalence of lung diseases in psoriasis and their underlying mechanisms can help implement appropriate preventative and therapeutic strategies to address respiratory diseases in patients with psoriasis.

## 1. Introduction

Psoriasis is a systemic inflammatory disease affecting 2–4% of the population that develops as a result of abnormal interactions between the innate and adaptive immune responses [1]. The classical manifestations of psoriasis include raised areas of inflamed skin covered with silvery-white scales, mainly over the extensor surfaces of the limbs, on the scalp, and within the lumbosacral region [2]. The disease can also affect the joints, causing psoriatic arthritis [3].

The systemic inflammation in psoriasis may be associated with the development of comorbidities [4]. Interestingly, concomitant conditions can already be detectable in children and adolescents with the disease [5]. Patients with psoriasis have an increased risk of hyperlipidemia [6], hypertension [7], diabetes [8], and an increased body mass index (BMI) [9]. Furthermore, psoriasis has been associated with a higher prevalence of gastrointestinal [10] disorders and chronic kidney disease [11]. 

Recent studies also suggest an increased incidence of lung diseases in psoriasis. The development of lung diseases in populations with psoriasis may be due to several mechanisms, including exposure to common risk factors, increased susceptibility to respiratory diseases in psoriasis, or other mechanisms (Figure 1).

Lung diseases adversely affect the quality of life of patients with psoriasis and contribute to the overall burden of disease. In this review, we aimed to provide a summary of the current evidence regarding the prevalence of lung diseases in patients with psoriasis and the potential underlying mechanisms.

## 2. Materials and Methods

We carried out a comprehensive review of PubMed- and EMBASE (Elsevier, Amsterdam, The Netherlands)-indexed articles in order to assess the prevalence of lung diseases in psoriasis, as well as to identify factors that could explain the seemingly higher prevalence of lung diseases in patients with psoriasis. This systematic review was conducted in accordance with the preferred reporting items for systematic reviews and meta-analyses (PRISMA) statement.

### 2.1. Search Strategy

A comprehensive review of PubMed- and EMBASE-indexed articles was carried out. Our literature search strategy included selected search terms combined using the following Boolean operators of “AND” and “OR”: (asthma) OR (chronic obstructive pulmonary disease) OR (obstructive sleep apnea) OR (pulmonary hypertension) OR (interstitial lung disease) OR (sarcoidosis) AND (psoriasis). Search terms were identified using the MeSH thesaurus. To minimize any potential bias, there were no limits on the language of publication. 

### 2.2. Selection Criteria 

The authors developed a list of studies during the selection process. The search was narrowed by adding a filter for publication dates between March 2010 and September 2021. We also obtained additional studies from the reference lists of relevant reviews. Subsequently, the references and abstracts of articles found on PubMed and EMBASE were screened to identify potentially relevant publications. Next, the full texts of the identified articles were retrieved, and the articles were reviewed to narrow the selection to relevant reports. This stage served as an in-depth verification of findings. Eligible studies included the following: clinical trials, observational studies, meta-analyses, systematic reviews, case series, and case reports published during the predefined period. The exclusion criteria included the following: serious weaknesses in study design, e.g., failure to choose a control group or the use of an inappropriate control group in studies that weren’t case reports/case series; articles where the full text was not retrievable; and articles where the outcomes could not be attributed to device.

Once the selection of the articles was completed and all the full texts of the selected articles were retrieved, data extraction followed. The methodology used in the literature search and review is detailed in Figure 2 and Figure 3.

### 2.3. Data Extraction

A total of 23 relevant publications published from March 2010–September 2021 were selected during the PubMed- and EMBASE-indexed literature search. The authors extracted data from the included studies using a standard form. Data on the following were extracted: (1) basic characteristics of the included studies (full reference, publication year, country, ethnicity, number of patients, age range, percentage of male patients, and lung disease); (2) prevalence rates of lung diseases in question. The domains would be rated for their appropriateness as yes, partly, unsure, or no, respectively.

## 3. Results

A total of 195 references were identified through electronic database searches and searched by title and abstract. Following the exclusion of duplicate and irrelevant references, a total of 128 articles were considered essential, reviewed in their full text, and selected or rejected based on the exclusion criteria. All articles were reviewed according to the scheme shown in Figure 2. A total of 23 original reports that met the inclusion and exclusion criteria were analyzed, including one case report, one case series, seventeen retrospective reviews, zero prospective studies, zero comparative studies, and four RCTs. Table 1 summarizes the studies included in the analysis. Of the 195 articles included in the review, 43 related to asthma, 18 to chronic obstructive pulmonary disease (COPD), 26 to obstructive sleep apnea (OSA), 3 to pulmonary atrial hypertension (PAH), 53 to interstitial lung disease (ILD), and 52 to sarcoidosis. 

## 4. Discussion

### 4.1. Prevalence of Lung Diseases in Psoriasis

A number of studies reported an increased prevalence of asthma [12], chronic obstructive pulmonary disease [20,21], obstructive sleep apnea [25], and pulmonary hypertension [31] in psoriasis patients. There are also single studies that report an increased prevalence of interstitial lung disease and sarcoidosis in psoriasis [32,34]. To our knowledge, ours is the first systematic review to discuss the prevalence and pathophysiology of lung diseases in psoriasis.

The risk of developing individual lung diseases in psoriasis varies with sex, age, and the severity of psoriasis. In terms of sex, a Taiwanese trusted source study found an increased risk of COPD in men with psoriasis (HR = 2.38, 95% CI: 1.36–4.18) [23]. There were no similar relationships found for other lung diseases. Older patients with psoriasis (≥50 years) had a higher risk of asthma (OR = 1.64, 95% CI: 1.44–1.88) and chronic obstructive pulmonary disease (HR = 2.19, 95% CI: 1.27–3.77) than younger ones [12,23]. In children, observational studies suggest an increased risk of asthma compared to the general population, but the data are inconsistent [13,17,18,19]. Furthermore, the increased susceptibility to asthma in children was associated with moderate to severe psoriasis [17]. 

In terms of psoriasis severity, it was shown that even those with mild psoriasis had an increased risk of developing COPD (OR = 1.66, 95% CI: 1.00–2.76) [21] and sarcoidosis (HR = 1.49, 95% CI: 1.18–1.87); however, moderate to severe psoriasis also showed an increased risk of developing COPD (OR = 2.15, 95% CI: 1.26–3.67) [21], pulmonary hypertension (OR = 1.46, 95% CI: 1.09–1.94) [31], and sarcoidosis (HR = 2.51, 95% CI: 1.64–3.85) [34]. An increased risk of pulmonary hypertension has not been observed in patients with mild psoriasis [31]. There was no association between psoriasis severity and an increased risk of asthma [12] or OSA [35]. 

Certain lung diseases, such as interstitial lung disease [32] and sarcoidosis [34], have an uncertain relationship with psoriasis. Interstitial lung disease was occasionally reported in patients with psoriasis, with most cases identified as immunosuppressant-induced pneumonitis [36,37,38,39,40,41]. In a study on patients with interstitial lung disease not treated with immunosuppressants, 4.7% were also diagnosed with psoriasis [32]. However, another study showed that pulmonary fibrosis was not more common in psoriasis than would be expected in the general population [42]. In one study, researchers found that people with psoriasis were more predisposed to developing sarcoidosis than controls [34]. Further large prospective studies are warranted to investigate the association between psoriasis and both interstitial lung disease as well as sarcoidosis.

The main issue with studies that assess the association between psoriasis and lung diseases seems to be their disregard of common risk factors associated with both respiratory problems and psoriasis. Failure to adjust for these factors may result in risk overestimation of the discussed lung diseases. Thus, additional well-designed studies are urgently needed to address the potential contribution of pulmonary risk factors to the development of lung diseases in patients with psoriasis.

### 4.2. Common Risk Factors

Common risk factors, responsible for both the development and worsening of lung diseases and psoriasis, include smoking, obesity and low physical activity, pollutants, infections, exposure to allergens, malnutrition, metabolic syndrome, connective tissue disorders, depression, and the use of certain medications (Figure 4). 

#### 4.2.1. Smoking

Research shows that patients with psoriasis report smoking, both passive and active, more often than the general population [63]. Smoking has been shown to significantly correlate with both the development and severity of psoriatic lesions [64]. At the same time, smoking is recognized as the most important causative factor in chronic obstructive pulmonary disease [43]. Other lung diseases that have been linked to smoking include asthma [44] and idiopathic pulmonary fibrosis [45]. Thus, smoking can be a factor that explains the increased prevalence of lung disease in patients with psoriasis.

#### 4.2.2. Obesity and Low Physical Activity

Obesity and low physical activity have been identified as risk factors for the development and severity of psoriasis, and a higher prevalence of obesity has been shown in patients with psoriasis [65,66]. Obesity is a principal risk factor for the development of respiratory diseases such as asthma [55], pulmonary hypertension, and sleep apnea [56]. It has also been implicated in chronic obstructive pulmonary disease [57]. Thus, obesity may explain the higher prevalence of lung disease in psoriasis.

#### 4.2.3. Pollutants

Cadmium, the element found in tooth fillings, used in battery and television manufacturing, as well as the aircraft industry, has been proposed as an air pollutant linked to the development of psoriasis. Higher levels of cadmium in the blood have been observed in patients with psoriasis [67]. At the same time, a higher blood cadmium level was associated with chronic obstructive pulmonary disease in males, including those who had never smoked [62]. Thus, cadmium (and potentially other air pollutants) can equally impact the development of psoriasis and lung diseases in the same individuals.

#### 4.2.4. Infections

Infections can play a role in both psoriasis and lung diseases. The altered microbiome in the lower respiratory tract, which can increase the risk of psoriasis, interacts with the innate mucosal immune system and has been implicated in the development of asthma [18,60]. Furthermore, human immunodeficiency virus (HIV) is a well-known risk factor associated with psoriasis and lung diseases such as asthma, chronic obstructive pulmonary disease, and pulmonary arterial hypertension [59,61,68]. It is possible that HIV infection contributes to the development or worsening of both psoriasis and lung diseases.

#### 4.2.5. Allergy

Patients with psoriasis have been shown to present weakly expressed hypersensitivity to some inhalant, food, and contact allergens, such as birch, artemisia, timothy grass and rye pollens, house dust mites, and molds [69]. The intensity of hypersensitivity reactions correlated with PASI [69]. Sensitization to allergens also plays an important role in the development, severity, and treatment of asthma. Recent studies have found that an allergy to house dust mites, pet dander, cockroaches, or fungi is a risk factor for asthma [58]. 

#### 4.2.6. Malnutrition

The composition of the intestinal microbiota can influence the susceptibility to an increased incidence of many inflammatory diseases. Body mass index and the consumption of red wine were shown to affect the intestinal microbial composition. Diet-induced dysbiosis of intestinal microbiota may induce cytokine imbalance, which is the underlying mechanism in psoriasis [70]. However, dysbiosis of the intestinal microbiota has also been implicated in asthma through early life antimicrobial exposure, Caesarian birth, formula feeding, lack of maternal exposure to pets or livestock during pregnancy, and maternal consumption of antimicrobials during pregnancy [51].

#### 4.2.7. Metabolic Syndrome and Its Components

A higher prevalence of hyperlipidemia [6], hypertension [7], and diabetes mellitus [8] was shown in patients with psoriasis. Hyperlipidemia is also associated with poorer hyperinflation and airway obstruction in patients with chronic obstructive pulmonary disease [52]. Obstructive sleep apnea is common in patients with hypertension (a prevalence rate of 37% to 56%), and the prevalence rate in those with resistant hypertension can be as high as 70% to 83% [53]. Individuals with diabetes are at an increased risk of the following several pulmonary conditions: asthma, COPD, PAH, and fibrosis [54]. Thus, the individual components of metabolic syndrome may explain the increased risk of lung disease in psoriasis.

#### 4.2.8. Connective Tissue Disorder

It has been accepted that patients with an autoimmune disease have a higher risk of developing another autoimmune disorder [71]. Whereas the data on the coexistence of psoriasis and systemic lupus erythematous are unclear [72], systemic sclerosis was found to be independently associated with psoriasis [73]. Furthermore, the leading causes of death in systemic sclerosis are pulmonary arterial hypertension and interstitial lung diseases [49,50]. Hence, connective tissue disorder, which shares an autoimmune etiology with psoriasis, may lead to lung disease in its late stages, thus explaining the link between psoriasis and lung disease.

#### 4.2.9. Depression

Psoriasis is a chronic inflammatory dermatosis that limits patients’ ability to work and socialize, affecting their family life as well as leisure and sexual activity [74]. These limitations can be a source of stress, which is a well-established trigger of psoriasis [75]. On the other hand, depression is a common and chronic comorbidity in patients with interstitial lung disease [48]. The use of antidepressants has been associated with a significantly increased risk of idiopathic pulmonary arterial hypertension [46]. Moreover, in people suffering from depression, previous studies have shown an increase in proinflammatory cytokines such as TNFα and IL-6, similar to that in psoriasis [76]. In this way, depression can be a link between psoriasis (which leads to depression) and lung disease (which may, for instance, develop in response to the treatment of depression with selective serotonin reuptake inhibitors).

#### 4.2.10. Medications

Apart from the previously mentioned SSRIs, which have been linked to idiopathic pulmonary arterial hypertension [46], there are medications that can induce bronchospasm, e.g., β-blockers, angiotensin-converting enzyme (ACE) inhibitors, nonsteroidal anti-inflammatory drugs (NSAIDs), and interferons [47]. Interestingly, many of them, that is, β-blockers, angiotensin-converting enzyme inhibitors, nonsteroidal anti-inflammatory drugs, lithium, anti-malarial drugs, interferons, imiquimod, terbinafine, tetracycline, and fibrate drugs [77], have been mentioned in the context of drug-related psoriasis, referring to the onset or exacerbation of psoriasis associated with certain medications. Thus, medications can contribute to the development or exacerbation of both psoriasis and certain lung diseases in patients prescribed the above treatments for other indications.

#### 4.2.11. Other Factors

There was a single report of erythrodermic pustular psoriasis induced by intravesical bacillus Calmette-Guérin immunotherapy in a patient with bladder cancer [78]. There was also a single report of granulomatous pneumonia as a complication of bacillus Calmette-Guérin immunotherapy [79]. These show that there may also be other common factors, poorly understood to date, that lung diseases and psoriasis have in common. Thus, further research is warranted. 

### 4.3. Chronic Inflammation as the Underlying Mechanism of the Development of Lung Diseases in Psoriasis

Systemic inflammation in psoriasis triggers immune changes that can lead to the development of comorbidities [80], and patients with an inflammatory autoimmune disease have a higher risk of developing another autoimmune disorder [71,81]. So far, no studies have been conducted to evaluate the role of immunological mechanisms and inflammatory mediators in the pathogenesis of lung disease in patients with psoriasis. However, elevated levels of many pro-inflammatory cytokines have been found in skin lesions as well as in blood, but only in moderate to severe psoriasis, as measured by the PASI (Psoriasis Area and Severity Index) and defined as more than 42% of the body surface area [82]. It is therefore possible that, due to its possible autoimmune component, psoriasis-related immune dysfunction could trigger an abnormal immune response in the lungs, acting de novo on the lung tissue and exacerbating the pre-existing inflammation caused by the underlying disease [83,84,85,86]. Tumor necrosis factor-α, dendritic cells, interleukin-1, -6, and -8, and T-cells all contribute substantially to the pathogenesis of psoriasis and increase the risk of other systemic diseases [87]. The increased levels of inflammatory markers, such as tumor necrosis factor-α (TNF-α), interleukin-6, and C-reactive protein, have indeed been found in the lung tissue of patients with psoriasis [88]. 

Lung diseases share many aspects with psoriasis (Figure 5). The similarities include immune processes, cytokine profiles, and immune cell types. The immune profile of psoriasis is mediated by T helper 1 (Th1) and Th17 cells, but also by Th22 and Th9 activation [89]. Interestingly, neutrophilic inflammation in severe or corticosteroid-resistant asthma is mediated by Th1 and Th17 cells, as well as other neutrophilic mediators [90]. The same cells accumulate in the lungs of patients with stable chronic obstructive pulmonary disease [91,92] or in active disease sites, forming non-keratinizing granulomas in patients with sarcoidosis [93]. Th17 cells were also implicated in one of the most common pathways leading to alveolitis in interstitial lung disease [94]. 

#### 4.3.1. Th1 Cells

Th1 cell activation leads to an increase in interferon γ (INF-γ) and tumor necrosis factor α (TNF-α) levels [95]. INF-γ- and TNF-α-mediated inflammatory signaling is the key process in psoriasis [96]. However, elevated levels of the same inflammatory cytokines have also been found in the sputum and bronchopulmonary lavage fluid of patients with chronic obstructive pulmonary disease [97,98], as well as in the serum of patients with obstructive sleep apnea [97,98,99,100]. There are also data to support the role of INF-γ and TNF-α in pulmonary emphysema [101,102]. Moreover, TNF-α may upregulate goblet cell hyperplasia, which has been implicated in a number of lung diseases [103]. The above evidence shows that Th1 cells, as common mediators of inflammation in both psoriasis and lung diseases, add to the explanation of their concomitance.

#### 4.3.2. Th17 Cells

Th17 cells produce IL-17A, -17F, and -22, which are all known as highly inflammatory cytokines that induce the keratinocyte activation and proliferation seen in psoriasis [76,85,86,104]. The increased activity of the same cytokines has been observed in the sputum of patients with asthma [105], in bronchoalveolar lavage fluid of patients with sarcoidosis [106], as well as in the serum of patients with obstructive sleep apnea [107,108,109] and sarcoidosis [106]. The IL-17-high asthma phenotype, with bronchial epithelial dysfunction and an upregulated inflammatory response, resembles that of psoriasis [110]. In the murine lung epithelium, IL-17 overexpression led to inflammation with a chronic obstructive pulmonary disease-like phenotype involving CD4 cell recruitment, mucus hypersecretion, small airway fibrosis, and chemokine expression [94,111,112]. Thus, Th17 cell involvement could add to the understanding of links between psoriasis and lung disease.

#### 4.3.3. Th22 Cells

Th22 cells are identified by the production of IL-22, which is one of the members of the IL-10 family [113]. IL-22 has been implicated in both psoriasis and several lung diseases. In psoriasis, it is known to promote keratinocyte activation and the formation of epidermal acanthosis [114,115]. Elevated levels of IL-22 were found in patients with T-cell-mediated lung diseases, such as interstitial lung disease [116]. IL-22 has also been shown to induce smoking-dependent airway remodeling and emphysema-like alveolar enlargement in chronic obstructive pulmonary disease [117], as well as the recruitment, activation, and proliferation of mononuclear cells, contributing to alveolitis and granuloma formation in sarcoidosis [118]. 

#### 4.3.4. Th9 Cells

Th9 cells are the most recently defined subset of T-helper cells. They produce IL-9, which might be involved in immune-mediated disease [119]. The exact role of Th9 cells in the pathogenesis of psoriasis and lung diseases is still poorly understood, which warrants further research. It is postulated that Th9 cells may participate in initiating and maintaining skin inflammation [120]. IL-9 overexpression in the lungs of transgenic mice appears to enhance airway inflammation and play a role in the early initiation of airway remodeling, most notably mucus production within the epithelium [121]. Thus, alongside other cell subsets, Th9 cells can help explain the link between psoriasis and lung diseases. 

#### 4.3.5. Other Immune Mechanisms

Other immune mechanisms implicated in the pathogenesis of psoriasis and lung diseases include transcription factors (NF-κB, HIF-1α) [104]; T-cell receptor and co-stimulatory molecules (e.g., CD3, CD8) [84,122]; T-cell cytokines (e.g., interferon-γ, I L-13, IL-17, and IL-23) [83,84] and other inflammatory cytokines (e.g., TGF-β, IL-12, IL-18, IL-22, VEGF, and TRAIL) [84,123,124,125,126,127,128], chemokines and receptors (e.g., CXCR1, CXCR2, and CXCR3) [84,129], adhesion molecules (e.g., ICAM-1, VCAM-1, E-selectin, and CD18) [84,127,128,130], and proteases (elastase, cathepsins, and matrix metalloproteinases) [84,131]. They are believed to contribute to pathological changes in the lungs of patients with psoriasis, which are shown in Figure 6. 

### 4.4. Other Postulated Causal Mechanisms

#### 4.4.1. Genetic Factors and Genetic Instability

A potential genetic link between psoriasis and asthma, chronic obstructive pulmonary disease, or sarcoidosis has been proven. Single-nucleotide polymorphism (SNP)-based genome-wide association studies (GWASs) have identified common variants determining susceptibility to psoriasis and asthma [132,133,134]. In a pathway analysis, genes that were differentially expressed in asthma patients with high levels of IL-17 were shared with genes reported as being altered in psoriasis [110]. Another confirmation of the shared genetic background between psoriasis and asthma is a study conducted on twins, which, using a bivariate variance components twin model, demonstrated an increased occurrence of asthma in patients with psoriasis [16]. Genetic analyses revealed a copy number variation of the β-defensin genes in both psoriasis [135] and chronic obstructive pulmonary disease [136]. Furthermore, the polymorphism in the promoter region of the TNF-alpha gene [137], previously established in psoriasis, was also found in patients with chronic obstructive pulmonary disease [138]. Sarcoidosis, alongside several other autoimmune disorders, including psoriasis, has been demonstrated to be associated with IL23R polymorphism [139,140]. All these findings support the genetic links between psoriasis and lung diseases.

When analyzing the common genetic pathways of psoriasis and lung diseases, the phenomenon of genetic instability should be mentioned, which can be observed as cytogenetic instability and/or molecular instability.

Cytogenetic instability mainly relates to chromosomal instability (CNI), with a high number of breaks and/or numerical and structural chromosome aberrations. DNA damage, hereinafter referred to as the genetic DNA fragment between sister chromatids, can be detected using the analysis of sister chromatid exchange (SCE) [141]. In one study, significantly higher SCE rates were observed in the peripheral lymphocytes of patients with psoriasis [142]. Variations in SCE have also been found in other chronic diseases, but COPD does not appear to be associated with DNA damage [143].

Molecular instability is associated with microsatellite instability (MSI)—as an alteration of allele length due to a change in the number of nucleotide repeats, and/or allelic instability, as a loss of heterozygosity (LOH)—the loss of one of two examined alleles. In addition, MSI and LOH have been reported in a number of human malignancies [141]. Additionally, MSI and LOH have been detected in asthma, COPD, sarcoidosis, and IPF [144,145,146,147]. In part of these studies, specific regions of chromosomes 3p, 5q, 6p, 13q, 14q, and 17q containing (a) genes for susceptibility to asthma and allergy, as well as (b) genes associated with an increased risk of developing irreversible airway obstruction were analyzed [145]. There are no studies on the occurrence of MSI in psoriasis. One study found that specific LOH-altered genetic loci on chromosomes 3p, 7p/q, and 8p are important in the development of psoriatic plaques, but recent research indicates that psoriasis is not related to LOH [148,149]. Based on these results, it has been hypothesized that cigarette-smoke-induced acquired somatic mutations are a major contributor to COPD. Another possible hypothesis may be that MSI reflects a defect in the DNA repair process caused by the oxidative stress associated with smoking, which leads to insufficient airway remodeling [150]. Smoking is believed to cause potentially irreversible genetic changes in epithelial cells [151]. It is possible that cigarette smoking, which exacerbates psoriasis, may cause similar changes in genetic instability.

#### 4.4.2. Epigenetic Mechanisms

Epigenetic processes represent potential molecular routes between genetic backgrounds and environmental risk factors contributing to the pathogenesis of psoriasis and pulmonary diseases. Recent studies highlight the role of epigenetic processes in inflammatory diseases, including psoriasis [152]. Similarly, evidence of aberrant epigenetic marks has emerged in patients with asthma, chronic obstructive pulmonary disease (COPD), and pulmonary arterial hypertension (PAH). Epigenetic mechanisms include the regulation of gene expression at the transcriptional (via DNA methylation and histone modification) and post-transcriptional levels (via long non-coding RNAs and microRNAs) [153].

There are numerous reports on the role of histone modification in psoriasis. Psoriasis is associated with the incorrect expression of histone acetyltransferases (HATs) and histone deacetylases (HDACs), which regulate the balance between histone acetylation and deacetylation. In patients with psoriasis, elevated mRNA levels of HDAC1 compared to healthy controls were noted [154]. Aberrant HDAC activities were also observed in pulmonary fibrosis [155]. Moreover, Ding et al. noted that the expression of HDAC1 and HDAC2 was upregulated in mice exposed to cigarette smoke, a common risk factor for psoriasis and lung diseases [156]. Other studies concerned the mechanisms of histone modulation in controlling cytokine production and its role in the onset of psoriasis and asthma. It has been found that among psoriasis patients, H3K4 methylation is increased in peripheral blood mononuclear cells [157]. The genome-wide mapping of histone modifications in CD4 memory as well as TH1 and TH2 cells revealed differential enrichment of dimethylated lysine 4 of histone H3 (H3K4me2) in TH2 enhancers associated with susceptibility to asthma [158]. It has also been noted that increased mRNA levels of DEFB1 (beta-defensin 1) are correlated with H3K4me3 in the progression of COPD [159].

MicroRNA (miRNA) plays an important role in the pathogenesis of psoriasis. In psoriasis-affected keratinocytes, the increased expression of miR-146a, miR-203, miR-21, miR-31, miR-184, miR-221, and miR-222 was observed, whereas the expression of miR-99a, miR-424, and miR-125b was decreased [160]. MiR-146a, along with miR-146b, participates in homeostasis and controls inflammatory responses in the skin [161]. Moreover, they are negative regulators of inflammatory gene expression in lung tissue [162]. The consistent upregulation of miR-146b was observed in a murine model study of acute and chronic asthma [163]. In the skin, the increase in miR-203 is consistent with the downregulation of suppressor of cytokine signaling-3 (SOCS-3) and the subsequent upregulation of signal transducer and activator of transcription-3 (STAT-3) [164,165]. The transcriptional downregulation of SOCS3 has been observed in asthma and COPD patients [166,167]. The expression of miR-21 is increased in skin affected by psoriasis due to T-cell infiltration [168]. MiR-21 has been associated with the activation of fibroblasts and their differentiation to myofibroblasts in the pathogenesis of asthma, PAH, and IPF [169,170,171]. The increased expression of miR-221 in psoriasis participates in the regulation of keratinocytes and immune cell proliferation [160]. miR-221 is relevant in controlling the aberrant proliferation of human airway smooth muscle cells in severe asthma [172]. In contrast to the above, miR-125b, responsible for proliferation, is downregulated in psoriatic lesions [21]. In another study, it has been noted that miR-125b in the sputum supernatant was reduced in COPD patients [173].

Another interesting case of post-translational modification concerns sphingosine-1-phosphate receptor-1 (S1PR1), a high-affinity G protein-coupled receptor expressed at the endothelial cell (EC) surface and relevant for the regulation of the immune system [174]. The S1P-S1PR pathway is involved in the pathogenesis of immune-mediated diseases, including psoriasis [175]. The post-translational modification of S1PR1 at serine and tyrosine-143 residues plays a role in disrupting endothelial barrier function, leading to inflammatory conditions of the lungs [176].

#### 4.4.3. Treatment of Psoriasis

Medications used for the treatment of psoriasis can sometimes trigger or exacerbate lung diseases, especially interstitial lung disease. This is an important consideration from the perspective of comorbidity. 

Methotrexate, cyclosporine, acitretin, and fumaric acid ester are systemic drugs commonly used in psoriasis. Even though methotrexate [36] and fumaric acid ester [38] have been reported to cause lung fibrosis, this is an extremely rare complication; therefore, all three can be considered generally safe for the respiratory system. Patients treated for psoriatic arthritis with other immunosuppressants, such as leflunomide [39] and sulfasalazine [41], have been reported to develop interstitial lung disease. 

Biological medications target specific parts of the immune system. Because they are immunosuppressants, they carry an increased risk of upper respiratory tract infections [177]. According to the latest data, biological drugs used in the treatment of psoriasis probably have a similar risk of serious infections or infections of the respiratory tract [178]. Biologic therapy has been proven to be efficacious for the treatment of psoriasis and psoriatic arthritis. These drugs also affect lung tissue. 

Asthma appears to be a definite but rare side effect of the anti-TNF blockade [179]. Moreover, anti-TNF causes exacerbations of asthma, bronchospasm, or wheezing, although most of these have been related to anaphylaxis. This is because it is likely that following the introduction of a TNF-α-blocking drug, the Th1 response was suppressed, allowing the clinical expression of the Th2 pathway as asthma [180]. The appearance of asthma has not been observed with the use of other biological drugs. In the past, anti-TNF-α therapy was viewed as a therapy that caused pulmonary fibrosis. Recent studies have shown that anti-TNF-α therapy does not increase the occurrence of ILD among patients with autoimmune diseases [181]. 

For the treatment of psoriasis, the following three IL-17-related monoclonal antibodies have been registered: secukinumab, ixekizumab, which directly blocks IL-17A, and brodalumab, which blocks IL-17A receptors [182]. There are no data on the negative effects of anti-IL-17 on lung function. On the contrary, IL-17-neutralizing therapy could be used to treat lung diseases such as asthma, COPD, and possibly fibrosis [183]. Recent studies suggest that asthma may be able to be separated into subtypes: Th2-dominated, Th17-dominated, and Th2/Th17-low, which suggests that αIL-17 therapy may prove effective in the treatment of some types of asthma [184]. Similarly, both COPD and IPF appear to be multidimensional diseases, suggesting that increased levels of IL-17 correlate with more severe disease progression [185,186]. 

Similar positive conclusions can be drawn from the use of anti-IL-23 in pulmonary disorders in mice models. Treatment with an anti-interleukin-23 antibody attenuated airway inflammation as well as fibrosis and reduced interleukin-17A and -22 levels in a murine model with the exacerbation of pulmonary fibrosis [187]. Treatment with an anti-IL-23p40 monoclonal antibody significantly weakened porcine-pancreatic-elastase-induced emphysematous changes in the lungs of wild-type mice. Targeting IL-23 in emphysema appears to be a therapeutic strategy for delaying disease progression [188].

#### 4.4.4. Underdiagnosis and Treatment of Lung Disease

The underestimated prevalence of lung diseases in psoriasis may result from a mild course of lung disease that remains underdiagnosed. The diagnosis may also be obscured by the presence or current treatment of another concomitant lung disease. An example is pulmonary hypertension, which can develop secondarily into chronic obstructive pulmonary disease, obstructive sleep apnea, sarcoidosis, and pulmonary fibrosis [189]. Whereas all conditions can occur in patients with psoriasis and their prevalence is higher than in the general population (which was discussed above), the presence of COPD, obstructive sleep apnea, sarcoidosis, or pulmonary fibrosis and its current treatment may obscure the clinical features of pulmonary hypertension, thereby leading to its underdiagnosis.

## 5. Conclusions

Patients with psoriasis have an increased risk of asthma, chronic obstructive pulmonary disease, obstructive sleep apnea, and pulmonary hypertension, which are important, previously unrecognized causes of morbidity and mortality. The development of lung diseases in patients with psoriasis can be explained by several mechanisms, including common risk factors (smoking, obesity, air pollution, infections, exposure to allergens, malnutrition, cardiovascular disease, connective tissue disorder, depression, and use of certain medications), shared immune and molecular characteristics associated with chronic inflammation, and other mechanisms. Understanding the prevalence of lung diseases in psoriasis and their underlying mechanisms can help to develop appropriate preventative and therapeutic strategies to address respiratory diseases in patients with psoriasis.

## Figures and Tables

**Figure 1 ijms-23-01767-f001:**
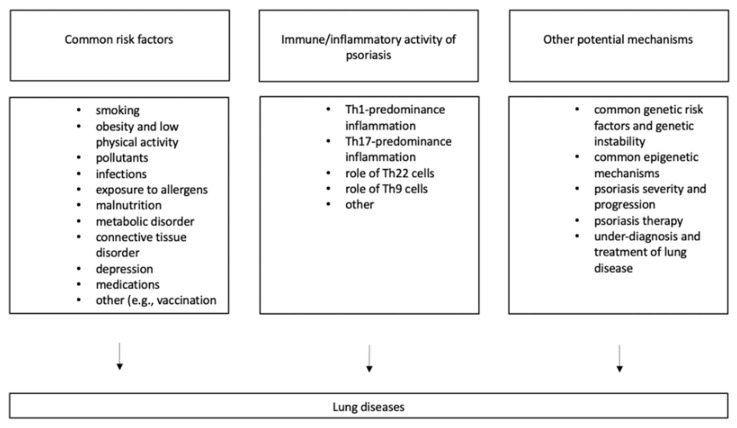
Potential mechanisms for the risk of lung diseases in psoriasis.

**Figure 2 ijms-23-01767-f002:**
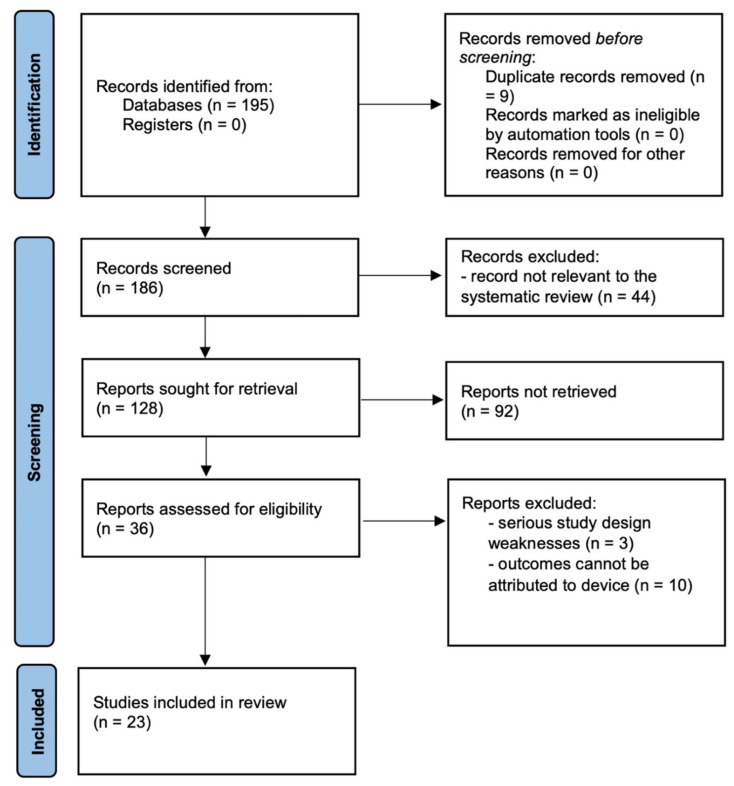
Literature search and review methodology.

**Figure 3 ijms-23-01767-f003:**
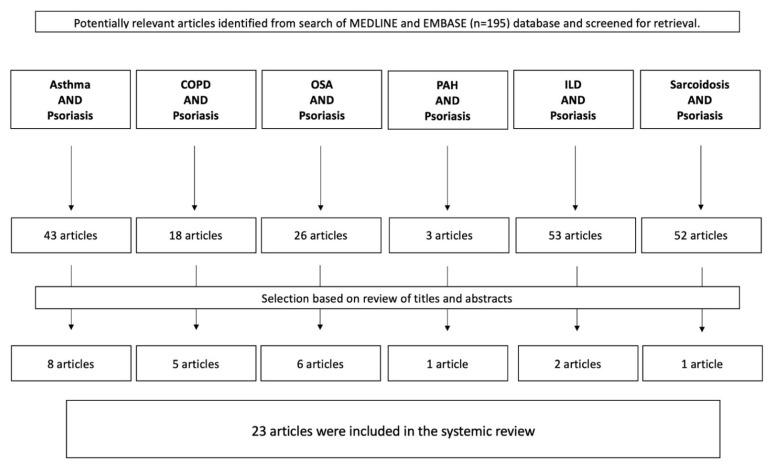
Literature selection process with subdivisions into selected lung diseases. Abbreviations: COPD, chronic obstructive pulmonary disease; OSA, obstructive sleep apnea; PAH, pulmonary atrial hypertension; ILD, interstitial lung disease.

**Figure 4 ijms-23-01767-f004:**
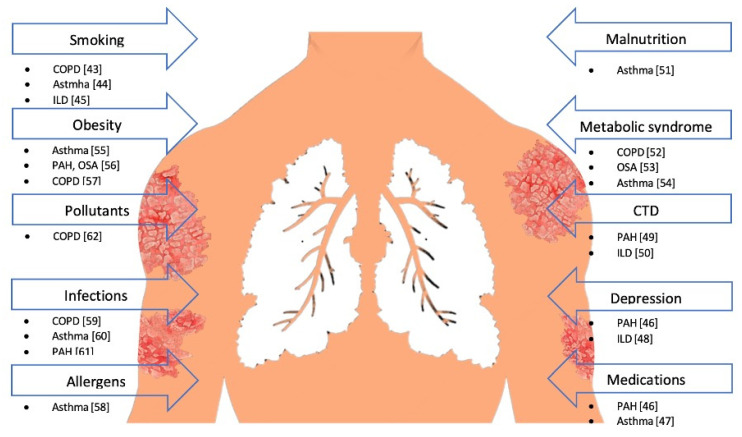
Risk factors common to psoriasis and lung diseases. Abbreviations: CTD, connective tissue disorder; COPD, chronic obstructive pulmonary disease; IPF, idiopathic pulmonary fibrosis; PAH, pulmonary atrial hypertension; and OSA, obstructive sleep apnea [43,44,45,46,47,48,49,50,51,52,53,54,55,56,57,58,59,60,61,62].

**Figure 5 ijms-23-01767-f005:**
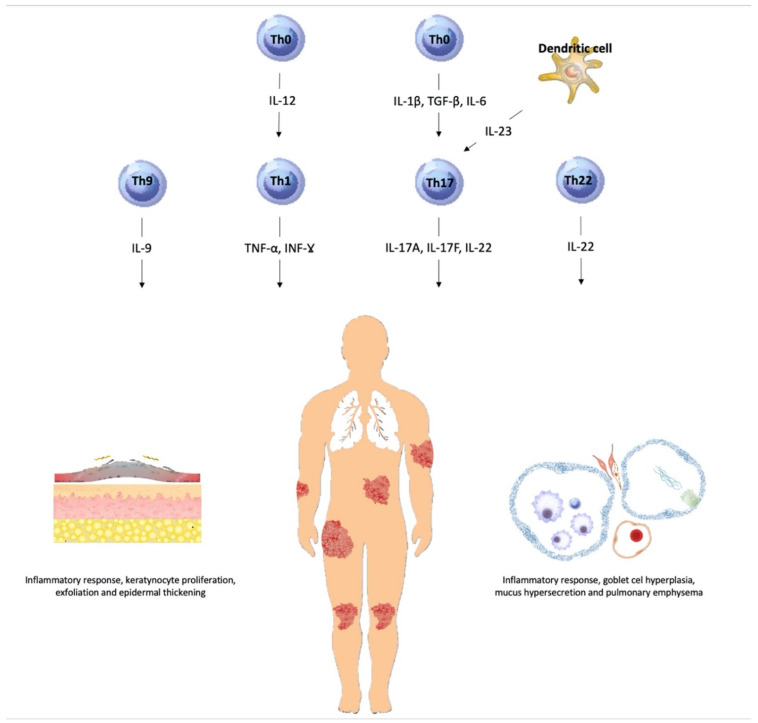
Immunological similarities between psoriasis and lung diseases. Abbreviations: Th, T helper cell; IL, interleukin [89,90,91,92,93,94].

**Figure 6 ijms-23-01767-f006:**
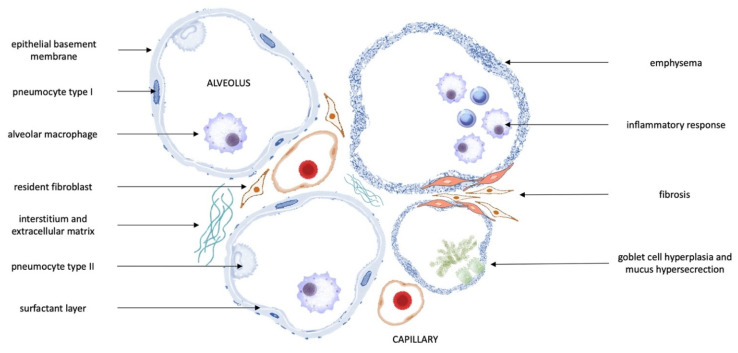
Potential abnormalities in the lungs of patients with psoriasis [94,101,102,103,111,112,117,118,120,121].

**Table 1 ijms-23-01767-t001:** Summary of studies included in the systematic review.

References	Study Design	Cases, No. with Lung Disease/Total No.	Controls, No. with Lung Disease/Total No.	Males, Case/Control, %	Patient Ages, Mean +/− SD, y	Age of Controls, Mean +/− SD, y	Prevalence Rate	Confounders Controlled for
Asthma								
Wang J et al. [12]	Meta-analysis	1713/66,772	34,220/577,415	N.A.	All ages, N.A.	All ages, N.A.	OR 1.32 (95% CI, 1.20–1.46)	Age, ethnicity
Augustin M et al. [13]	Retrospective cohort study	160/1313	27,319/291,868	N.A.	<18, N.A.	<18, N.A.	PR 1.34 (95% CI, 1.14–1.59)	N.A.
Fang HY et al. [14]	Retrospective cohort study	420/10,288	1153/41,152	52.7/52.7	≥20, 43.5 ± 17.0	≥20, 43.2 ± 17.1	HR 1.38 (95% CI, 1.23-1.54)	Age, sex, and comorbidities
Hajdarbegovic E et al. [15]	Cross-sectional study	19/301	14/147	N.A./44	All ages, N.A.	All ages, 54 ± 15.6	OR 1.60 (95% CI, 0.77–3.32)	N.A.
Lønnberg AS et al. [16]	Cross-sectional study	151/1385	2,714/31,993	N.A.	≥20	≥20	OR 1.32 (95% CI, 1.11–1.57)	Sex, age, smoking, body mass index, and chronic obstructive pulmonary disease
Galili E et al. [17]	Cross-sectional study	345/3112	70,363/884,653	N.A.	16–18, N.A.	16–18, N.A.	OR 1.44 (95% CI, 1.29–1.61)	Age, sex, BMI, socio-economic status, country of origin, and number of siblings
Kim SY et al. [18]	Retrospective cohort study	196/325	63,751/164,963	44.1/44.1	<15	<15	IR 3.94 (95% CI, 2.16 to 7.17)	Age, sex, income, region of residence, and comorbidity
Egeberg A et al. [19]	Retrospective cohort study	87/6586	21,638/1,456,385	N.A.	6–14, N.A.	6–14, N.A.	IRR 3.85 (95% CI, 2.15–6.90)	N.A.
Chronic Obstructive Pulmonary Disease								
Ungprasert P et al. [20]	Meta-analysis	N.A./331,347	N.A.	N.A.	N.A.	N.A.	OR 1.45 (95% CI, 1.21–1.73)	N.A.
Li X et al. [21]	Meta-analysis	6673/42,150	14,368/163,174	N.A.	N.A.	N.A.	OR 1.90 (95% CI, 1.36–2.65)	N.A.
Wu CY et al. [22]	Case-control study	N.A./1127	N.A./1127	46.3/46.3	N.A., 53.1	N.A., 52.5	OR 1.68 (95% CI, 1.02–2.77)	None
Chiang YY et al. [23]	Retrospective cohort study	25/2071	42/8342	N.A.	N.A.	N.A.	HR 2.43 (95% CI, 1.48–3.98)	N.A.
Al-Mutairi N et al. [24]	Retrospective case-control study	98/1835	74/1835	52.5/52.5	N.A., 52.3 ± 11.9	N.A., 52.7 ± 13.5	OR 1.46 (95% CI, 1.06–2.01)	Drug, smoking status
Obstructive Sleep Apnea								
Ger TY et al. [25]	Meta-analysis	N.A.	N.A.	N.A.	N.A.	N.A.	OR 2.60 (95% CI, 1.07–6.32)	Age, sex, and body mass index
Saçmacı H et al. [26]	Cross-sectional study	18/60	4/60	50/50	N.A., 42.8 ± 13.1	N.A., 43.6 ± 13.9	OR 6 (95% CI, 1.89–19.04)	N.A.
Papadavid E et al. [27]	Case-control study	24/27	229/330	79.2/79	>18, 50.7 ± 13.6	>18, 53 ± 11.8	OR 13.31 (95% CI, 1.19–48.93)	Age, sex, overweight/obesity, central obesity, comorbidity, and smoking status
Shalom G et al. [28]	Case-control study	327/12,336	369/24,008	52.2/50.4	>20, 55.6 ± 16.3	>20, 54.0 ± 17.1	OR 1.27, 95% CI, 1.08–1.49	Age, sex, ethnicity, body mass index, and comorbidity
Papadavid E et al. [29]	Cross-sectional study	19/35	N.A.	65.7/N.A.	N.A., 48.9 ± 13.06	N.A.	N.A.	Age, sex, body mass index, comorbidity, and smoking status
Yang YW et al. [30]	Cohort study	11/2258	25/11,255	62.6/62.6	18–59, N.A.	18–59, N.A.	30 (95% CI = 1.13–4.69)	Patients’ monthly incomes, geographic location, urbanization level, and obesity
Pulmonary Atrial Hypertension								
Choi YM et al. [31]	Retrospective cohort study	221/13,936	817/69,360	48.1/48.1	N.A. 57.1 ± 14.86 in mild psoriasis group, 52.4 ± 13.68 in severe psoriasis group	N.A. 57.6 ± 14.84 in mild psoriasis group, 52.9 ± 13.67 in severe psoriasis group	1.25 (1.05–1.49) in mild psoriasis group, 1.55 (1.16–2.06) in severe psoriasis group	Age, sex, comorbidity, and medications
Interstitial Lung Disease								
Ishikawa G et al. [32]	Case series study	21/N.A.	426/N.A.	66.7/N.A.	≥20, 66 ± 20	≥20, N.A.	N.A.	Age, sex, comorbidity, body mass index, smoking, and medications
Gupta R et al. [33]	Case report	N.A.	N.A.	N.A.	N.A.	N.A.	N.A.	N.A.
Sarcoidosis								
Khalid U et al. [34]	Cohort study	100/70,125	9717/5,973,393	48.3/49.5	≥10, 42.2 ± 18.3 in mild psoriasis group, 41.0 ± 16.6 in severe psoriasis group	≥10, 37.0 ± 21.8	IR 1.18 (CI, 1.15–1.20)	Age, sex, comorbidity, concomitant medications, and socioeconomic status

N.A. = not available/not applicable; OR = odds ratio; PR = prevalence ratio; HR = hazard ratio; IR = incidence ratio; and IRR = incidence rate ratio.

## Data Availability

The data presented in this study are available in this article.

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
