# Peer review of "Chronic Inflammation as the Underlying Mechanism of the Development of Lung Diseases in Psoriasis: A Systematic Review"

_ijms, 2022, doi:10.3390/ijms23031767_

Round 1
Reviewer 1 Report
The manuscript entitled “Chronic Inflammation as the Underlying Mechanism of Lung Diseases Development in Psoriasis: a Systematic Review” by Mateusz Mleczko et al, describes the existing data in context current evidence regarding the prevalence of lungs disease in patients with psoriasis and also discusses the related potential mechanisms. The authors of this article addressed the importance of the findings in terms of the prevalence of asthma, chronic obstructive pulmonary disease, interstitial lung disease, obstructive sleep apnea, pulmonary hypertension and sarcoidosis in psoriasis.
The authors have given reported several studies in the manuscript based on the available literature, but there are certain sections that need to be elaborated and some needs the addition of literature.
General comments:
- Authors have discussed number of risk factors, for example, Obesity and low physical activity, pollutants, infections, Malnutrition, etc
- Authors have discussed role of various cell types, but I wonder, if authors would discuss some epigenetic factors as well
- Role of pos-translational modifications in lung diseases that may be the causative agents or initiation factors towards the development of psoriasis
- I think authors should also investigate some genetic factors and MSIs based on the different populations, viz, European, Asian and others.
- Various studies describing pos-translational modifications in lung diseases, viz; Anwar and Mehta, 2020 etc., those studies should be discussed.
- Role of mono-clonal antibodies to be delivered and to be used as specific immunotherapeutic.
Author Response
Dear Reviewer,
We would like to thank you for careful and thorough reading of this manuscript and for the comments and constructive suggestions, which help to improve the quality of this manuscript. As below, on behalf of my co-authors, I would like to clarify some of the points raised in this review.
Comment 1:
Authors have discussed number of risk factors, for example, Obesity and low physical activity, pollutants, infections, Malnutrition, etc
We greatly appreciate the Reviewer for this valuable comment. We would like to kindly inform that our intention was to summarize data from Figure 4. We hope this form will be sufficient.
Comment 2:
Authors have discussed role of various cell types, but I wonder, if authors would discuss some epigenetic factors as well.
Response:
We thank the Reviewer for this opinion. In fact, epigenetic processes that represent potential molecular routes between genetic back-grounds and environmental risk factors contributing to the pathogenesis of psoriasis and pulmonary diseases. Epigenetic mechanisms that involve the regulation of gene expression at the transcriptional and post-transcriptional levels, are presented in a separate paragraph in accordance with the remarks (see “Epigenetic mechanisms”).
Comment 3:
Role of pos-translational modifications in lung diseases that may be the causative agents or initiation factors towards the development of psoriasis.
Response:
Thank you very much for your appropriate point. Indeed, we have observed that miRNAs play an important role in the pathogenesis of psoriasis. Moreover, some of these miRNAs are negative regulators of the expression of inflammatory genes in lung tissue. We have summarized our observations in the “Epigenetic mechanism” section.
Comment 4:
I think authors should also investigate some genetic factors and MSIs based on the different populations, viz, European, Asian and others.
Response:
This is a very important point and we thank you for it. Until now, we were not aware of the importance of MSIs in the pathogenesis of psoriasis and lung disease. Therefore, we extended the paragraph "Genetic factors" to include the phenomenon of genetic instability. In terms of individual populations, we found no significant differences. Many of the sources used in this systematic review relate to the pathogenesis of psoriasis and lung disease in specific populations, but the data are inconsistent (see Table 1).
Comment 5:
Various studies describing pos-translational modifications in lung diseases, viz; Anwar and Mehta, 2020 etc., those studies should be discussed.
Response:
This is very interesting case of post-translational modifications in lung diseases. Thank you for your comment. It turns out that the S1P-S1PR pathway is also involved in the pathogenesis of immune-mediated diseases, including psoriasis. For more information, see the "Epigenetic mechanisms" section.
Comment 6:
Role of mono-clonal antibodies to be delivered and to be used as specific immunotherapeutic.
Response:
We appreciate this comment. Biologic therapy has been proven to be efficacious for the treatment of psoriasis and psoriatic arthritis. These drugs also affect lung tissue. This influence can be manifold. We extended the "Treatment of psoriasis" chapter to include the effects of anti-TNF, anti-IL17 and anti-IL23 drugs on lung tissue. The use of these drugs in the treatment of psoriasis may have important clinical implications in terms of lung disease.
Is the English used correct and readable? Moderate English changes required
Response:
We thank the Reviewer for drawing attention to the need of correction of English grammar and flow. We checked manuscript by MDPI Author Services and corrected all indicated errors. We hope that the quality of English grammar in revised manuscript will be satisfactory for the Reviewer.
Reviewer 2 Report
This is an interesting systematic review assessing the association of pulmonary disease and psoriasis. The format of the table makes it a little difficult to read (letters are running together, etc.) The models are appropriate and helpful. I suggest that the authors add a section discussing current medications approved for treatment of psoriasis and how/if they affect the pulmonary system. I think that will improve the quality of the manuscript.
Author Response
Dear Reviewer,
We would like to thank you for careful and thorough reading of this manuscript and for the comments and constructive suggestions, which help to improve the quality of this manuscript. As below, on behalf of my co-authors, I would like to clarify some of the points raised in this review.
Comment 1:
This is an interesting systematic review assessing the association of pulmonary disease and psoriasis.
Response:
We thank the Reviewer for this opinion. To our knowledge, ours is the first systematic review to discuss the prevalence and pathophysiology of lung diseases in psoriasis. We hope this study will contribute to broadening the knowledge in this field.
Comment 2:
The format of the table makes it a little difficult to read (letters are running together, etc.) The models are appropriate and helpful.
Response:
We greatly appreciate the Reviewer for this valuable comment. We agree that Table 1 may be too extensive. This was due to great number of clinical data that need to be differentiated. In accordance with the comment, we changed the layout and removed redundant content. We hope this form will be sufficient.
Comment 3:
I suggest that the authors add a section discussing current medications approved for treatment of psoriasis and how/if they affect the pulmonary system. I think that will improve the quality of the manuscript.
Response:
The influence of biological drugs in the treatment of psoriasis is a very important point and we thank you for it. Biologic therapy has been proven to be efficacious for the treatment of psoriasis and psoriatic arthritis. These drugs also affect lung tissue. This influence can be manifold. We extended the "Treatment of psoriasis" chapter to include the effects of anti-TNF, anti-IL17 and anti-IL23 drugs on lung tissue. The use of these drugs in the treatment of psoriasis may have important clinical implications in terms of lung disease.
Round 2
Reviewer 1 Report
The revised manuscript entitled “Chronic Inflammation as the Underlying Mechanism of Lung Diseases Development in Psoriasis: a Systematic Review” by Mateusz Mleczko et al, describes the existing data in context current evidence regarding the prevalence of lungs disease in patients with psoriasis and also discusses the related potential mechanisms.
The authors have reported several number of studies in the manuscript based on the available literature, thus contributed genuinely to the growing field after the revision.
Author Response
Dear Reviewer,
We would like to thank you for careful and thorough reading, and the insightful review of this manuscript. We are very pleased that our article has met your expectations and does not require any additional corrections.
Comment 1:
The authors have reported several number of studies in the manuscript based on the available literature, thus contributed genuinely to the growing field after the revision.
Response:
We thank the Reviewer for this opinion. To our knowledge, ours is the first systematic review to discuss the prevalence and pathophysiology of lung diseases in psoriasis. We hope this study will contribute to broadening the knowledge in this field.